# Significant Substrate Effects on Electromagnetic Scattering by Particles in the Infrared Atmospheric Window

**Feifei Gao** [1], **Shangyu Zhang** [1,2], **Wenjie Zhang** [1,2,*], **Lanxin Ma** [1,2] **and Linhua Liu** [1,2]

1   School of Energy and Power Engineering, Shandong University, Jinan 250061, China
2   Optics & Thermal Radiation Research Center, Shandong University, Qingdao 266237, China
*   Correspondence: zhangwenjie@sdu.edu.cn

**Abstract:** Particle-dispersed coatings emerged as a promising approach to regulate the apparent radiative properties of underlying substrates in various applications, including but not limited to radiative cooling, thermal management, and infrared stealth. However, most research efforts in this field overlooked the dependent scattering mechanisms between the particles and the substrate, which can impact the optical properties of the particles. In this study, we explored the particle-substrate interactions within the atmospheric radiative window of 8–14 μm. Using the T-matrix method, we calculated the scattering and absorption efficiencies of a dielectric/metallic particle situated above a metallic/dielectric substrate, considering the different gap sizes. Near the small gaps ($<0.5a$ with $a$ the sphere radius), we found that the strong local fields induced by the interaction between the induced and image charges largely enhanced the absorption and scattering efficiencies of the particles. With the increasing gap sizes, the absorption and scattering efficiencies presented a significant oscillation with a period of about $4.5a$, which was attributed to the interference (standing wave) between the scattered fields from the sphere and the reflected fields from the substrate. Our findings identify a crucial role of the particle–substrate interactions in the infrared properties of particles, which may guide a comprehensive insight on the apparent radiative properties of the particle composite coatings.

**Keywords:** particles on substrate; electromagnetic scattering; localized surface phonon-polariton resonances; local field enhancements; standing waves

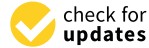



## 1. Introduction

Particle composite coatings acquired a lot of research interest in diversified applications relating to radiative heat transfer, including but not limited to radiative cooling [1–4], thermal control [5], and infrared stealth [6–9]. In these realistic applications, the coatings are painted on the substrate to regulate the apparent emissivity of objects, by well designing their electromagnetic responses. Hence, it is of vital importance to build efficient electromagnetic models to design and optimize the desired radiative properties. Still, the numerically exact Maxwell solvers of the coatings are hindered by the limited computational resources and large time consumption due to the countless particles and complex configurations [10,11]. In engineering fields, the radiative transfer equation (RTE) derived as immediate corollaries of Maxwell equations can feasibly predict the radiative properties of particle-dispersed coatings [12–16].

To solve RTE, the absorption and scattering properties of single particle must be first evaluated by the electromagnetic scattering theory. In radiative calculations, the pigment particles are usually assumed to be spherical, which gives excellent results due to averaging over many particles [17]. The scattering of a spherical particle can be described by the Lorent–Mie theory, which only considers the particle material, size, and coating matrix [18]. However, the dispersed particles may be deposited near the boundary between the coating and substrate because of gravity and polymer diffusion [19,20]. Recently, although the gradient distributed particles due to the sedimentation were considered in

the radiative cooling coating, the near-boundary particles were still described by the Mie theory in infinite space which neglects the particle-substrate interactions [21]. Until now, the dependent scattering between the particle and substrate was rarely considered in particle composite coatings.

Hence, two important questions arise naturally: (i) what is the influence of a substrate on the electromagnetic scattering by particles, and (ii) can the influence be averaged out or safely neglected in the particles dispersed systems? To answer the first question, many theoretical and numerical methods are extended including the T-matrix method [22–27], transfer matrix method [28], discrete dipole approximation method [29–31], finite-difference time-domain method [32–34], finite element method [35], and boundary element method [36,37]. Based upon these methods, the problem (i) was much studied for various particle-on-substrate couples, such as the couples of metal-on-metal [38,39], metal-on-dielectric [40,41], dielectric-on-metal [42,43], dielectric-on-dielectric [29,44], and semiconductor-on-substrate [45–48]. Current studies focused mainly on the substrate modified surface modes in the visible, near-infrared, and microwave bands. Despite a recent mid-infrared study, it only reported the absorption properties of a dielectric particle influenced by the touching substrate [49]. In this result, the influence mechanism of the substrate on scattering by particles remains ambiguous in the mid-infrared regime, which is significantly important for applications of the coatings in the atmospheric window.

The answer to the second question relies closely on the way the particle-substrate interactions evolve in a large range of gap sizes. Until now, the knowledges are based mainly on the small gap sizes that are smaller than the diameter of the particle (or even zero gap size for a touching substrate) [39–41,49–52]. Recently, Mackowski [26] found that the scattering cross sections of a particle significantly oscillate with the distances of the particle from the substrate. This oscillation can be attributed to the interference between the backscattered field from the sphere and the forward scattering field that is reflected from the substrate [26], which is also known as the standing wave [53]. Therefore, it is imperative to provide a detailed illustration of the oscillations induced by the interference (standing wave) at different gap scales.

In this work, using the multiple sphere T-matrix (MSTM) method and open-source code, we investigate the effects of gap sizes on the radiative properties of single particle above the substrate. We select two typical cases in applications which are a $SiO_2$ particle on Al substrate and an Al particle on $SiO_2$ substrate. In Section 2, we introduce the theoretical models of MSTM and verify the calculating models with a specific example of a gold sphere on glass. Additionally, in Section 3, we calculate the scattering and absorption cross sections of the particles for the two cases. We identify two different mechanisms that dominate over the different range of gap sizes. At small gaps, the intense local-field enhancement due to the induced charges enhance largely the absorption and scattering cross sections for the first and second cases, respectively. When gap scale increases, the standing waves induce a periodic oscillation on the optical cross sections. Hence, we reveal that the dependent scattering between the particle and substrate strongly modifies the scattering and absorption properties of the particles and should be further considered in practical applications. The conclusions are given in Section 4.

## 2. Materials and Methods

The $SiO_2$ and Al particles commonly used in coatings have widespread prospects for radiative properties regulation. $SiO_2$ particles are used directly or in mixtures as the primary substance for particle scattering in radiative cooling coatings [54–56]. Al particles are often used as materials for infrared stealth and thermal control coatings with low solar absorption and low infrared emission [8,15]. Both two types of particles settle in coatings and have different distances from substrates after curing. Moreover, in the prediction models of the radiation properties of particle composite coatings based on RTE, the dispersed particles are usually idealized as spherical particles. For convenience, we give the schematic diagram of the two models, as shown in Figure 1.

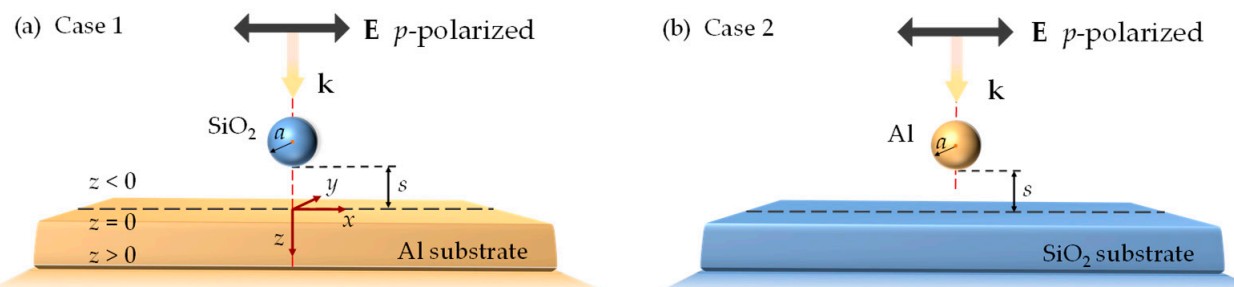

**Figure 1.** Schematic diagram of two geometric models studied in this paper. (**a**) Case 1: single SiO$_2$ spherical particle placed on the Al substrate; (**b**) case 2: single Al spherical particle located on the SiO$_2$ substrate. The coordinate system is shown, where $z = 0$ is set at the substrate surface. The radius of the sphere is $a$, while the gap size is represented by $s$. The *p*-polarized light is normally incident on the substrates with the incident wave vector **k**.

As seen in Figure 1, $s$ denotes the distance between the bottom of single sphere and the boundary of the substrate and $a$ is the radius of the spherical particle. The incident light is normally incident upon the substrate, which is along the positive direction of $z$-axis. The substrate surface is at $z = 0$, which separates the space into two layers. The particle exists in $z < 0$ that is denoted as $l = 0$ layer, while the substrate situates within the $l = 1$ layer. Note that the materials in this paper were isotropic and homogenous. The normally incident wave upon the substrate is considered, with the wave vector **k** is along the $+z$ direction. The *p*-polarized incident wave is studied in this work.

It was recently reported that MSTM is an accurate and efficient method to solve the scattering problem of the spherical particles near plane boundaries [26]. We used the open source program MSTM version 4.0 to calculate the scattering and absorption properties of the particles in the above models [57]. Conventional MSTM method gives the multiple spheres superposition solution of the frequency domain Maxwell's equations. The scattered field of each sphere has an associated multipole expansion form and further produces an exciting field on other spheres, which can be described by a multiple Green's function. When the substrate exists, the multiple Green's function is influenced by Fresnel relations at the plane boundary. Based on this process, Mackowski [26] recently extended MSTM to include the substrate through the plane boundary Green's boundary. Therefore, the extended MSTM method can solve the problem of single particle situated above the substrate.

The plane boundary Green's function $G^{\sigma\sigma'}(k_\rho, z, z')$ describes the amplitude of the wave in $\sigma$ direction at the target position $z$, which is caused by the wave emitted in the $\sigma'$ direction at the source position $z'$. $\sigma$ indicates the direction of wave propagation ($\sigma = +1$ and $-1$ indicate the positive and negative directions of the $z$-axis, respectively). $G^{\sigma\sigma'}(k_\rho, z, z')$ can be divided into indirect and direct parts [26]:

$$G^{\sigma\sigma'}(k_\rho, z, z') = \widetilde{G}^{\sigma\sigma'}(k_\rho, z, z') + \delta_{\sigma-\sigma'}\delta_{l-l'}\delta_{\text{sign}(z-z')-\sigma} \exp[i\sigma k_{z,l}(z-z')], \tag{1}$$

where the indirect term $\widetilde{G}^{\sigma\sigma'}(k_\rho, z, z')$ includes the influence of the boundaries, $k_\rho$ and $k_z$ represent, respectively, the lateral and normal wavenumbers, $l$ and $l'$ stand for, respectively, the layers containing $z$ and $z'$, and $\delta_n$ is the Kronecker delta function.

The *p*-polarized components of the scattered wave at a point $(r, \theta_{sca}, \phi_{sca})$ on the hemisphere in layer 0, which arises from the *q*-polarized excitation field, can be given by [26]

$$E_{sca,pq}(r, \theta_{sca}, \phi_{sca}) = \frac{1}{r}\exp(im_0 r)A_{pq}^-(k_{\rho,sca}, \phi_{sca}), \tag{2}$$

where $A_{pq}^{-}(k_{\rho,sca}, \phi_{sca})$ is the amplitude matrix of the scattered wave propagating along the negative direction of $z$-axis, $m_0$ is the complex refractive index of layer 0, and $q = 1, 2$ is the incident light parallel or perpendicular polarization.

Once the amplitude matrix of $A_{pq}^{-}(k_{\rho,sca}, \phi_{sca})$ in Equation (2) is determined, the extinction and scattering cross sections of the particle can be obtained. Since the extinction occurs for both positive and negative propagating waves in the presence of the substrate, the total far-field extinction cross section of the particle can be expressed as [26]

$$C_{f-ext} = C_{f-ext}^{+1} + C_{f-ext}^{-1} \tag{3}$$

$$C_{f-ext}^{\sigma} = -2\pi\mathrm{Re}\left[\sum_{q=1}^{2}\left(G_q^{\sigma\sigma_{exi}}(k_{\rho,exi}, Z_{\sigma B}, Z_{exi})\right) * A_{qp}^{\sigma}(k_{\rho,exi}, \phi_{exi})\right] \tag{4}$$

where $C_{f-ext}^{+1}$ and $C_{f-ext}^{-1}$ denote the reduction in incident wave energy due to the particle in the negative ($-1$) and positive ($+1$) directions along the $z$-axis, respectively. *f-ext* stands for the "far-field extinction". The extinction cross sections are defined such that the total monochromatic extinction power is $I_0 C_{f-ext}/k_0^2$, where $I_0$ is the excited plane wave intensity.

The total far-field scattering cross section are also composed of two parts. Each part is obtained by integrating the scattering flux over the hemisphere [26]:

$$C_{f-sca} = C_{f-sca}^{+1} + C_{f-sca}^{-1} \tag{5}$$

$$C_{f-sca}^{\sigma} = \int_0^{2\pi}\int_0^1\sum_{q=1}^{2}\left|A_{qp}^{\sigma}(\sin\theta_{sca}, \phi_{sca})\right|^2 d\cos\theta_{sca}d\phi_{sca} \tag{6}$$

The total far-field absorption cross section can be determined by

$$C_{f-abs} = C_{f-ext} - C_{f-sca}. \tag{7}$$

Thus, the scattering and absorption efficiencies are normalized by the geometric cross-sectional area of the sphere:

$$Q_{\mathrm{sca}} = \frac{C_{f-sca}}{\pi a^2} \tag{8}$$

$$Q_{\mathrm{abs}} = \frac{C_{f-abs}}{\pi a^2} \tag{9}$$

To verify the accuracy of the procedure in this work, we used the MSTM code to calculate the electromagnetic properties of metallic and dielectric spheres. In Figure 2a, the absorption efficiencies of a gold sphere above the glass substrate are shown, where the detailed parameters are given in the inset. Our results coincided well with the data calculated from the boundary element method (BEM) in Ref. [36]. In Figure 2b, we calculated the scattering efficiencies of a SiO$_2$ sphere in free space using the MSTM method and analytical Mie theory. The calculated results agree well between the two methods. The verification demonstrates a reliable calculation of the MSTM code used in this work.

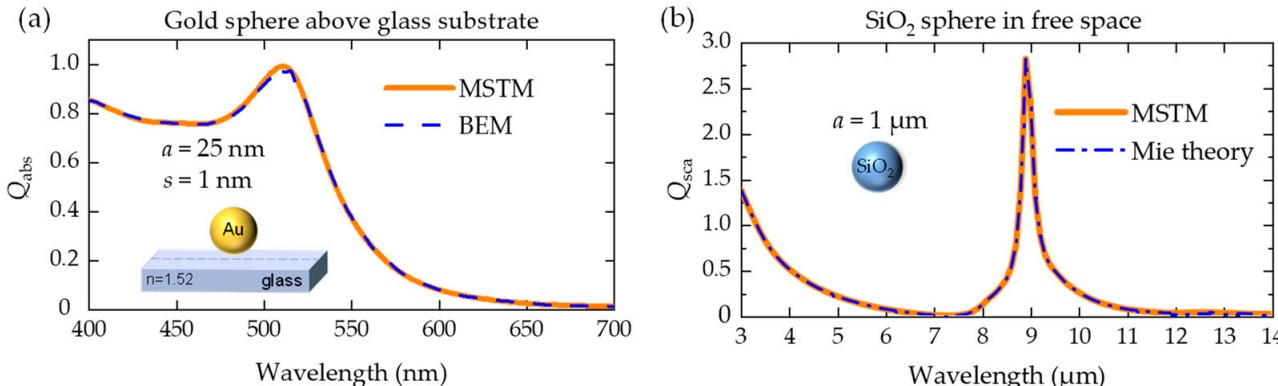

**Figure 2.** (**a**) The absorption cross sections of a gold sphere above the glass substrate calculated by MSTM and BEM [36]. (**b**) The infrared scattering efficiencies of a SiO$_2$ sphere in a vacuum are calculated by the MSTM method and Mie theory.

## 3. Results and Discussions

We calculated the scattering and absorption efficiencies for the two cases to understand the substrate induced regulations on the polar dielectric and metal particles. Figure 3a,b show the dielectric functions of SiO$_2$ [58] and Al [59], respectively. It was seen that the real part of SiO$_2$ dielectric function was negative within the range of 8.05–9.31 μm, which is known as the "Reststrahlen" band between the transverse and longitudinal optical phonon frequencies [60]. Optical constants of Al in the infrared band were dominated by the Drude model, which is clearly seen from Figure 3b. For convenience, the reflectance of the semi-infinite substrates is shown in Figure 3c. The reflectance of Al was almost 1 over the range of 8–14 μm, which was due to the impedance mismatch between Al and air. The reflectance of SiO$_2$ obviously presents a dispersive structure. Specifically, the reflectance of SiO$_2$ was lower than 0.1 in the spectral ranges of 8.1–8.9 μm and 11–14 μm, achieved the relatively high values between 0.1 and 0.4 within 8.9–11 μm, and peaks at about 9.2 μm with the value of 0.4.

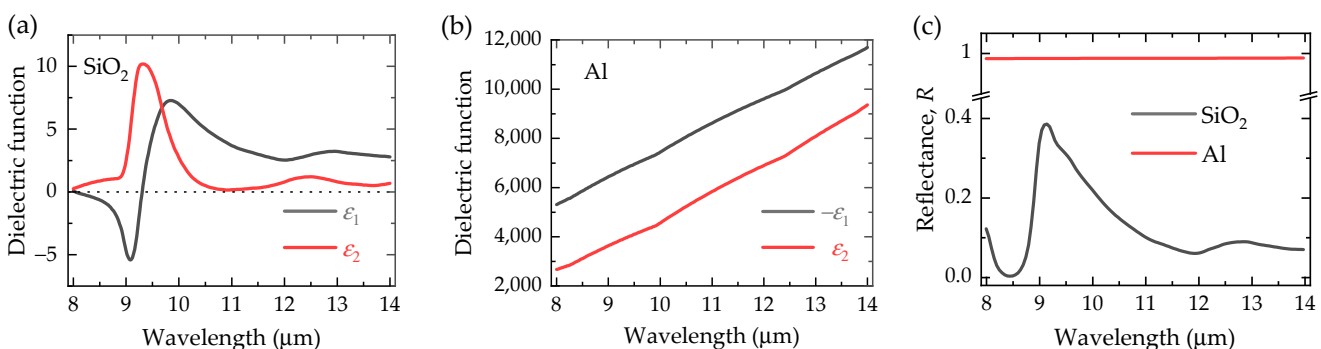

**Figure 3.** The dielectric functions of (**a**) SiO$_2$ and (**b**) Al. It is noted that the complex dielectric function is represented as $\varepsilon = \varepsilon_1 + i\varepsilon_2$. (**c**) The normal reflectance of semi-infinite Al and SiO$_2$ substrates. The spectral range is selected for 8–14 μm.

### 3.1. Case 1: Single SiO$_2$ Sphere above Al Substrate

For comparison, we first studied the optical properties of the single particle in the free space. In this case, the dependent scattering lacks in the absence of the substrate. The first rows of Figure 4a and b illustrate, respectively, the scattering $Q_{\mathrm{sca},0}$ and absorption efficiencies $Q_{\mathrm{abs},0}$ of the SiO$_2$ sphere in the vacuum or air, respectively. The radius of the sphere $a$ was 1 μm, which is a typical value in practical applications. It can be seen easily from the figure that the resonance structures exist in the scattering and absorption

efficiencies. According to the Fröhlich resonance conditions, the dipole resonance of a sphere in air occurs at

$$\varepsilon_1 \approx -2 - \frac{12}{5}q^2, \tag{10}$$

where $\varepsilon_1$ is the real part of the particle dielectric function and $q = 2\pi a/\lambda$ is the size parameter. After a fast calculation from Equation (10), the resonance occurs at about $\lambda = 8.9$ μm, which is consistent with the peak position in Figure 4. Since this Fröhlich resonance of SiO$_2$ spheres excites strong light–matter interactions within the atmosphere window, the realistic applications such as radiative cooling and regulation focus on this resonance structure a lot. In this regard, the substrate effects on the resonance should be inspected further in the following.

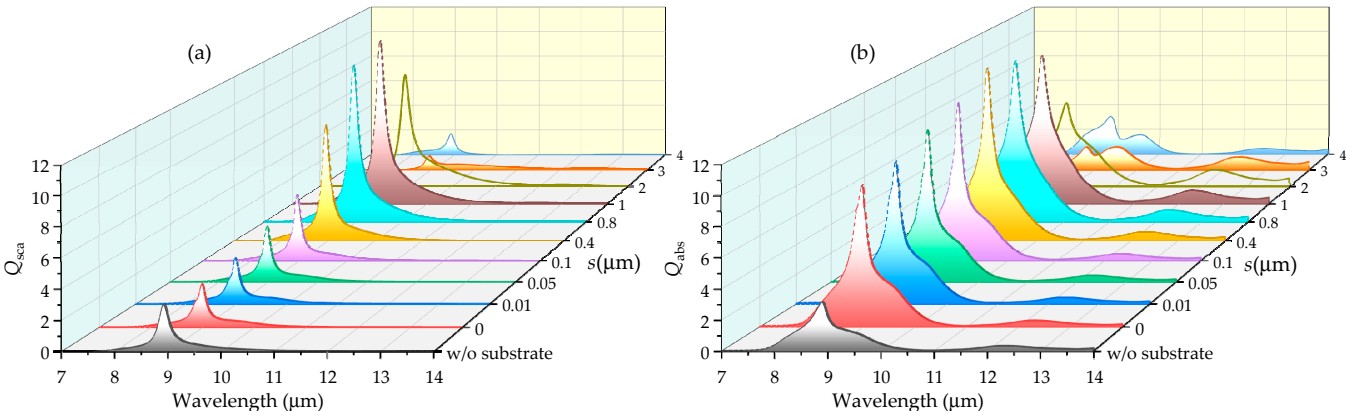

**Figure 4.** Mid-infrared properties of a SiO$_2$ sphere on the Al substrate versus the wavelength and particle-to-substrate distance *s, where curves of different colors correspond to different distances s of particles to the substrate*: (**a**) scattering and (**b**) absorption efficiencies. For convenience, the efficiencies of a free particle without the substrate $Q_{sca,0}$ and $Q_{abs,0}$ are also shown. The radius of the SiO$_2$ sphere is 1 μm.

Figure 4 also illustrates the scattering $Q_{sca}$ and absorption efficiencies $Q_{abs}$ of the SiO$_2$ sphere on the Al substrate, where the different rows correspond to the different particle-to-substrate distances *s*. It was clearly observed that both efficiencies are evidently regulated by the distance *s*. For scattering efficiencies, $Q_{sca}$ with the touching substrate (i.e., *s* = 0 μm) changes hardly compared with the free-space case $Q_{sca,0}$. With the increasing distance *s* from 0 to 0.1 μm, the scattering efficiencies increased slightly. However, the further increase in the distance from *s* = 0.1 μm to 1 μm will significantly enhance peak values of the scattering efficiencies. Specifically, $Q_{sca}$ for *s* = 1 μm present a near-4-fold increase compared with $Q_{sca,0}$ in free space. When the distance *s* continued to increase from 1 μm to 4 μm, $Q_{sca}$ decreased rapidly, as seen from Figure 4a. It is intriguing that $Q_{sca}$ fell below $Q_{sca,0}$ in free space for *s* = 3 μm and 4 μm.

As for absorption efficiencies in Figure 4b, the situation is somehow different from the scatting properties. The touching substrate (*s* = 0 μm) directly prompted the largely enhanced absorption efficiencies $Q_{abs}$ over the free-space case $Q_{abs,0}$, whereas the same enhancement in scattering requires a moderate distance (*s* ≈ 1 μm). This substrate-induced enhancement in $Q_{abs}$ lasts until the distance *s* is larger than 1 μm. Additionally, then $Q_{abs}$ suffers a decrease when *s* increases from 1 μm to 4 μm.

The enhanced mechanism of $Q_{abs}$ by the touching substrate arises from the increased local field enhancement which breaks the unitary limit achievable by a dipole resonance in vacuum [49,61,62]. The substantial enhancement of the local field between the sphere and substrate results from the near-field interaction of induced charges within the particle with their image in the metallic substrate mirror [49]. This enhanced mechanism of absorption results naturally in two following inspirations:

- The local fields-induced absorption enhancement could be traced from the electric field distribution near the particle;
- The local fields-induced absorption enhancement should vanish while the local fields enhancements disappear gradually with the moderately increasing distance.

Figure 5 illustrates the electric fields near the SiO$_2$ particle in the free space and above the substrate. The touching case ($s = 0$ μm) shows the intense enhancement of local fields in the gap, which agrees with the absorption enhancement in Figure 4b. However, when $s$ increased to be 0.4 μm and 1 μm, the largely vanishing local fields fail to explain the significant absorption enhancements for the corresponding distances in Figure 4b. As a result, in addition to the above local fields mechanism, there should be other mechanisms responsible for the substrate participated absorption.

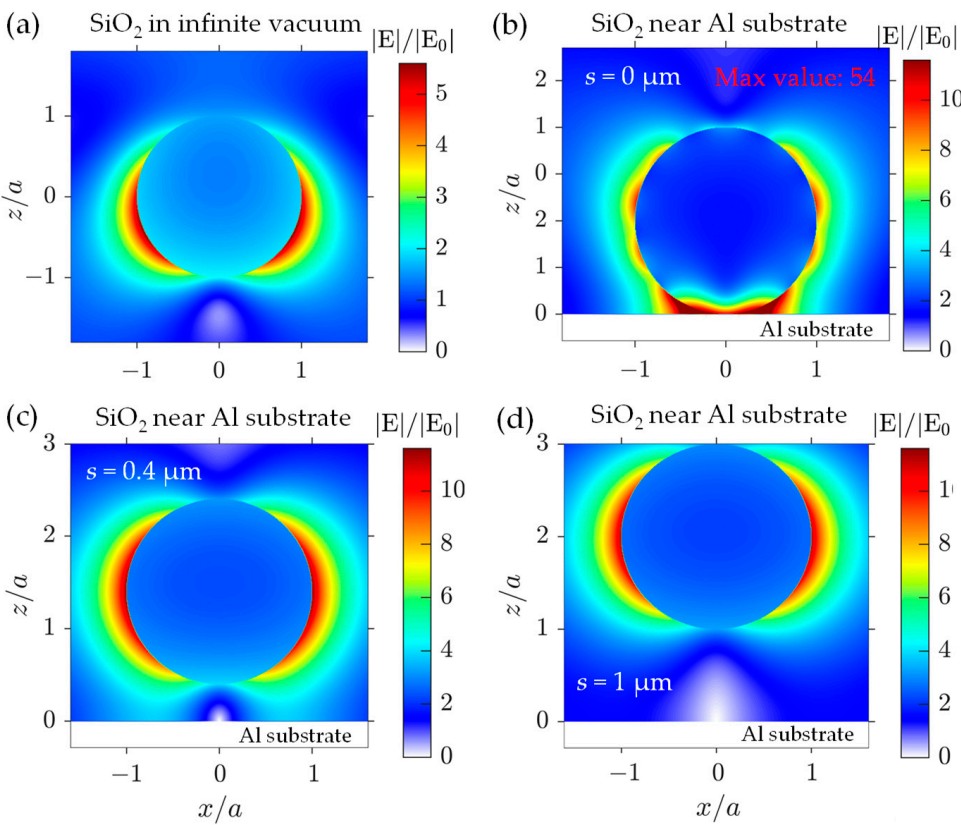

**Figure 5.** Normalized electric fields | E | / | E$_0$ | in *xz*-plane of a SiO$_2$ particle with $a = 1$ μm, where | E$_0$ | is the amplitude of the incident electric field. The incident wavelength is selected at the resonance position of $\lambda = 8.9$ μm. (**a**) The electric fields of the SiO$_2$ sphere in the free space. (**b–d**) The electric fields of the SiO$_2$ sphere above Al substrate with the varied distances of (**b**) $s = 0$ μm, (**c**) $s = 0.4$ μm, and (**d**) $s = 1$ μm. It should be noted that the maximum number of normalized fields in (**b**) is 54.

According to the high reflectance of Al substrate in Figure 3b, the incident wave was almost reflected by the substrate and then re-absorbed and re-scattered by the particle. When the particle was assumed to be absent, the interference occurred between the single-reflected waves upon substrate and the incident wave [24]. Additionally, in the presence of the particle, the interference was even more complex since the particle scattered waves were reflected by the substrate and then re-scattered by the particle. This process refers to the multiple scattering between the particle and substrate [26,53]. Multiple scattering process can be ignored when the particle is far away from the boundary. In this case, the total electric field **E** can be approximately as [53]

$$\mathbf{E} \approx \mathbf{E}_i + R\mathbf{E}_i + S(\mathbf{E}_i + R\mathbf{E}_i), \tag{11}$$

where $\mathbf{E}_i$ is the incident field, $R$ and $S$ represents the substrate reflecting and the particle scattering, respectively. In the regard, the first two terms in the right-hand side represent the interference between the incident and single-reflected waves. The latter two terms stand for the standing wave field (interference) between the directly scattering field $S\mathbf{E}_i$ and the secondly scattering field $SR\mathbf{E}_i$.

Therefore, there are two main mechanisms mediating the optical properties of particle above the substrate, *i.e.*, the induced charges at the small gap and the standing waves for the large distance. Essentially, the standing waves roots in the interference interaction which owns periodicity. To illustrate the two mechanisms, we calculated the optical properties in a broader range of distances $s$ from 0 μm to 12 μm. To further isolate the wavelength-dependent dispersion, we averaged the optical efficiencies of the $SiO_2$ sphere over the spectral range of 8–14 μm, which were termed as $Q_{sca}^{avg}$, $Q_{abs}^{avg}$ with the substrate and $Q_{sca,0}^{avg}$, $Q_{abs,0}^{avg}$ in the free space. The superscript "avg" represents the wavelength average.

In this regard, we calculated the ratios of $Q_{sca}^{avg}/Q_{sca,0}^{avg}$ and $Q_{abs}^{avg}/Q_{abs,0}^{avg}$ as seen in Figure 6. The touching substrate strongly enhanced the absorption within the particle, which arose from the interaction between the induced and image charges. With the increased distance, however, it was intriguing to observe a periodic fluctuation of the normalized efficiencies, which was dominated by the role of standing waves. According to Refs. [25,26,63], the period of the oscillating far-field cross sections can be analyzed by the phase term of $\exp(-2ikd\cos\vartheta)$, where $\vartheta$ is the incident angle, $k = 2\pi/\lambda$ is the wavenumber, and $d = a + s$. Therefore, the optical properties will oscillate with the period of $k(a + s)\cos\vartheta = \pi$ [26]. Taking the wavelength at the averaged value of 11 μm, we theoretically calculated the period of the distance $\Delta s \approx 4.5$ μm, which agreed well with the numerical results in Figure 6. It should also be noted that the oscillated amplitudes damp with the increasing distance, which implies a gradually weakening role of the standing waves on the particle.

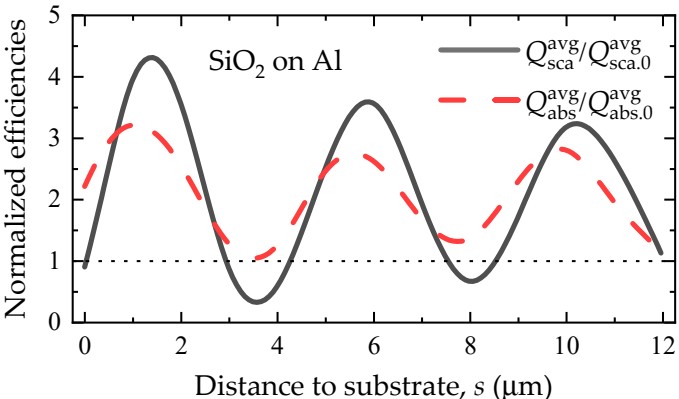

**Figure 6.** The wavelength-averaged efficiencies of the $SiO_2$ particle on the Al substrate versus the particle-to-substrate distance $s$. Both the scattering $Q_{sca}^{avg}$ and absorption efficiencies $Q_{abs}^{avg}$ are averaged over the spectral range of 8–14 μm. Note that the averaged efficiencies with the substrate are normalized with those without the substrate $Q_{sca,0}^{avg}$, $Q_{abs,0}^{avg}$.

The strongly regulated efficiencies by the Al substrate suggest that the replacements of $Q_{sca}$, $Q_{abs}$ with $Q_{sca,0}$, $Q_{abs,0}$ should be careful within the featured $s$ range in practical applications. For example, both average efficiencies considering the substrate suppress those in the free space over the almost whole range of $s = 0$–12 μm. Thus, even if the realistic $SiO_2$ particles distributes randomly over the Al substrate, a distance-averaged efficiencies of the particle system still deviate largely from those in free space. This implies that the effects of the substrate should be further considered in the radiative property prediction by solving RTE in the particle-dispersed coatings.

### 3.2. Case 2: Single Al Sphere above SiO2 Substrate

Figure 7 shows the scattering and absorption efficiencies of the Al sphere above the SiO2 substrate. The radius of the sphere was 1 μm and the spectral range was from 7 to 14 μm. We first focused on the isolated Al particle without the substrate. In this case, the absorption efficiency was much smaller than 1, which was due to the large dielectric function of Al as seen in Figure 3b. In the Rayleigh approximation, the dominant dipole absorption of the particle in the free space is given by

$$Q_{\text{abs},0} \approx q \frac{12\varepsilon_2}{(\varepsilon_1 + 2)^2 + \varepsilon_2^2}, \tag{12}$$

where $\varepsilon_1$ and $\varepsilon_2$ represent the real and imaginary parts of the particle dielectric function, respectively. The squares of the dielectric function in the denominator produce the very small absorption within the Al particle in the infrared range. Furthermore, the scattering efficiencies $Q_{\text{sca},0}$ can be about 100-fold larger than the absorption ones $Q_{\text{abs},0}$. Still, neither the absorption nor scattering efficiencies present any resonance structures for the isolated Al particle in the free space.

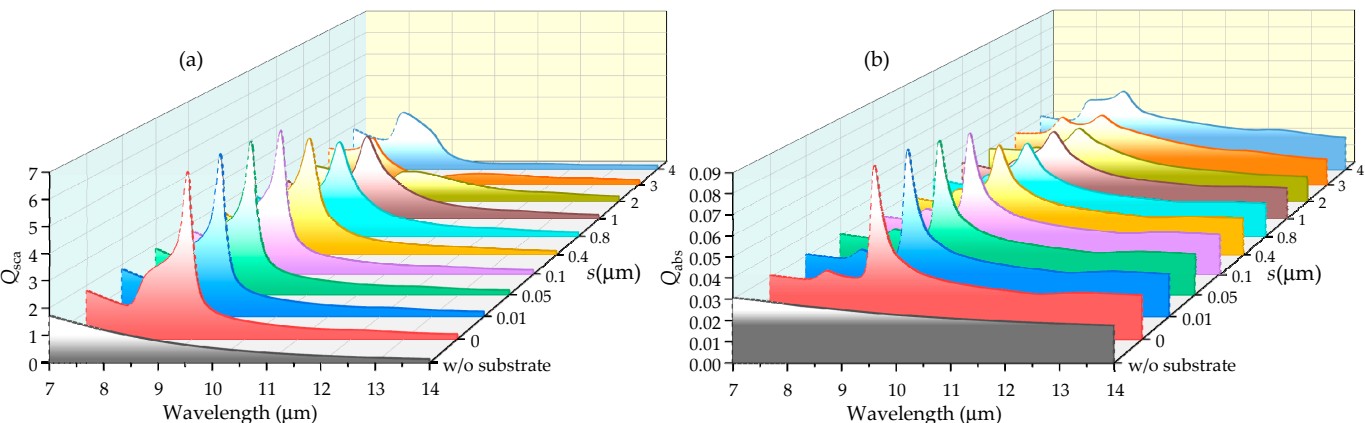

**Figure 7.** Infrared properties of an Al sphere on the SiO2 substrate versus the wavelength and particle-to-substrate distance *s, where curves of different colors correspond to different distances s of particles to the substrate*: (**a**) scattering and (**b**) absorption efficiencies. For convenience, the efficiencies of a free particle without the substrate are also shown. The radius of the Al sphere is 1 μm.

As for the Al spheres on the substrate ($s = 0$ μm), the resonance structures emerge based on the original plain lines. The position of the emerged resonances lies at about $\lambda \approx 9$ μm, which corresponds to the peaked wavelength of the SiO2 reflectance. At the peaked wavelength of about 9 μm, $Q_{\text{sca}}$ for the touching substrate reaches 6.4 that is nearly 7-fold higher than the isolated case $Q_{\text{sca},0}$. For absorption, $Q_{\text{abs}}$ for $s = 0$ μm obtains a 2-fold increase compared with $Q_{\text{abs},0}$. These suddenly enhanced scattering and absorption efficiencies may arise from the induced charges, which is explained in Section 3.1. However, the induced charges in this case enhance the scattering efficiencies largely rather than the absorption ones in Case 1.

With a slight increase in the distance *s* from 0 μm to 0.1 μm, the efficiencies decrease slightly implying the significant role of the induced charges for the small gap. When the distance increases further to 2 μm, the efficiencies suffer a large decrease which is due to the disappearing induced charges. As *s* increases from 2 μm to 4 μm, the scattering and absorption efficiencies increase again but hardly reach as large as those at small gaps ($s < 0.1$ μm).

To give an overall trend in a broad range of distances, the ratios of the wavelength averaged efficiencies above the substrate to those in the free space, $Q_{\text{sca}}^{\text{avg}}/Q_{\text{sca},0}^{\text{avg}}$ and $Q_{\text{abs}}^{\text{avg}}/Q_{\text{abs},0}^{\text{avg}}$, are plotted in Figure 8. Since the dielectric function and reflectance of SiO2 are strongly

dispersive over the 8–14 µm band, the wavelength average serves to highlight the influence mechanism of the particle-substrate distance on the optical properties of the particle. The first thing to note is that the substrate will enhance the wavelength averaged scattering and absorption efficiencies according to the results of $Q_{\text{sca}}^{\text{avg}}/Q_{\text{sca},0}^{\text{avg}} \geq 1$ and $Q_{\text{abs}}^{\text{avg}}/Q_{\text{abs},0}^{\text{avg}} > 1$. Second, the small gaps ($s < 1$ µm) can prompt a 2-fold enhancement on the averaged scattering efficiencies, whereas $Q_{\text{sca}}^{\text{avg}}/Q_{\text{sca},0}^{\text{avg}}$ drops rapidly under 1.5 with the further increasing $s$. Third, the oscillations in the averaged efficiencies can be roughly seen from the curves, although they are not as clear as in Case 1. The dispersion in the reflectance of $SiO_2$ may hinder the oscillation of standing waves which are closely related to the substrate reflectance according to Equation (11). After a rough estimation from Figure 8, the periods of the averaged scattering and absorption efficiencies were about 4 µm and 5 µm, respectively.

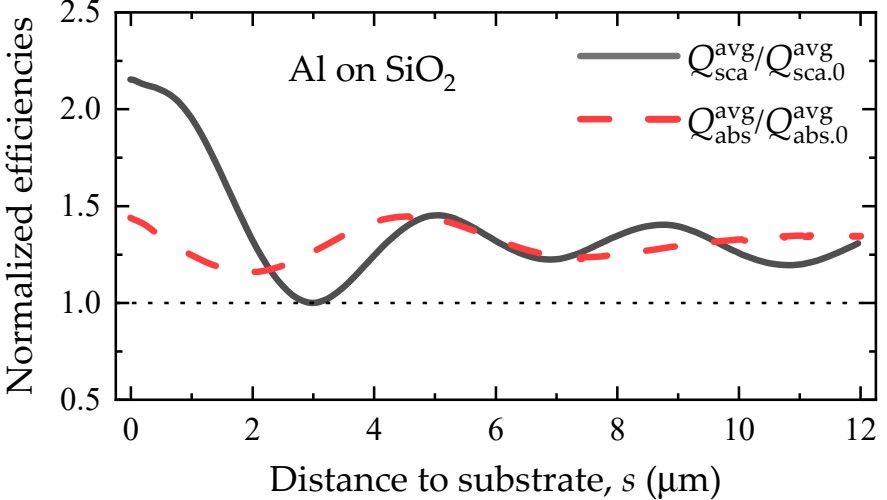

**Figure 8.** The wavelength-averaged efficiencies of the Al particle on the $SiO_2$ substrate versus the particle-to-substrate distance $s$. Both the scattering $Q_{\text{sca}}^{\text{avg}}$ and absorption efficiencies $Q_{\text{abs}}^{\text{avg}}$ are averaged over the spectral range of 8–14 µm. Note that the averaged efficiencies with the substrate are normalized with those without the substrate $Q_{\text{sca},0}^{\text{avg}}$, $Q_{\text{abs},0}^{\text{avg}}$.

Figure 9 shows the normalized electric fields in the *xz*-plane for single Al particle with and without $SiO_2$ substrate at $\lambda = 9$ µm. Figure 9a gives a typically non-resonant dipole pattern corresponding to an isolated metallic particle, where the fields distribute outside the particle and hardly penetrate the interior. In Figure 9b, the largely enhanced local field emerges near the particle-substrate junction, which illustrates the intense interaction between the induced and image charges. Furthermore, as seen from Figure 9b, many fields penetrate the $SiO_2$ substrate rather than the Al particle. This corresponds to the substrate induced scattering enhancement in Figure 8. It is imperative to note in this case that the enhanced local fields by the touching substrate induce a scattering enhancement, which is different from the absorption enhancement in Case 1. However, we also note the common characteristic between the two cases, i.e., the substrate enhanced fields can penetrate the polar material rather than the high-impedance metallic material. With the enlarged gaps in Figure 9c,d, the distributed pattern of the fields converts back into that of the isolated particle, but the amplitude of fields is twice over that in Figure 9a. In this regard, the field enhancements at the enlarged gaps should result from the interference between the particle scattering and substrate reflectance, which can be qualitatively analyzed from Equation (11).

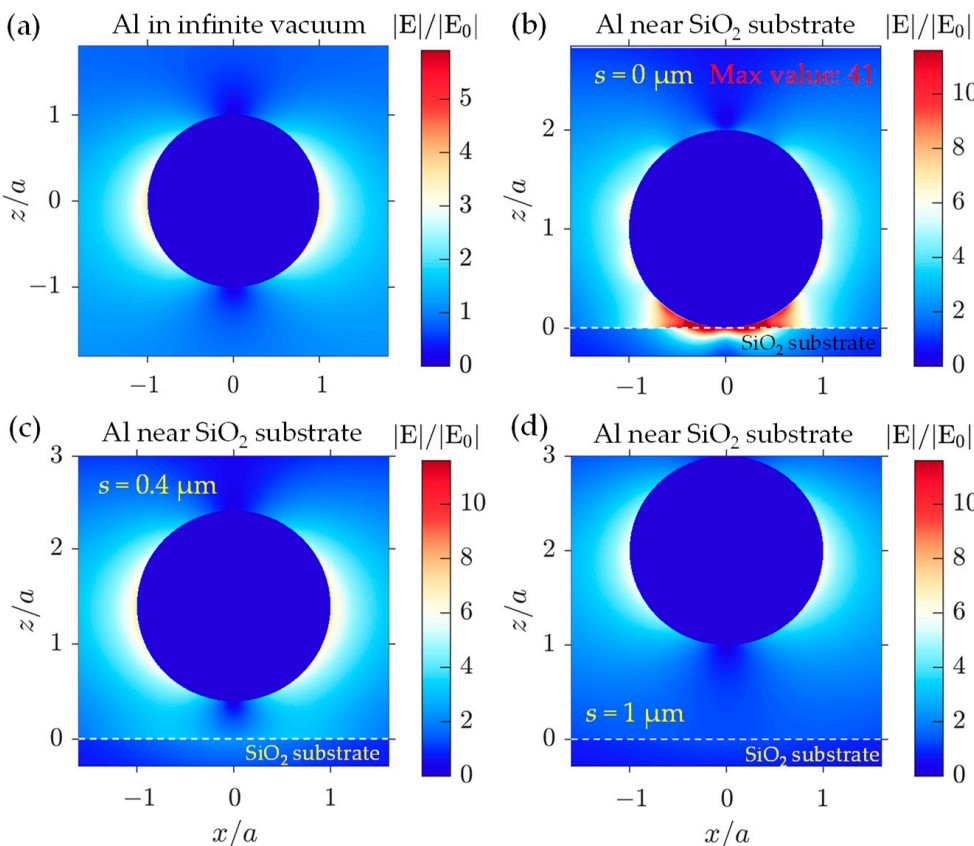

**Figure 9.** Normalized electric fields $|E|/|E_0|$ in $xz$-plane of an Al particle with $a = 1$ μm, where $|E_0|$ is the amplitude of the incident electric field. The incident wavelength is selected at $\lambda = 9$ μm. (**a**) The electric fields of the Al sphere in the free space. (**b–d**) The electric fields of the Al sphere above SiO$_2$ substrate with the varied distances of (**b**) $s = 0$ μm, (**c**) $s = 0.4$ μm, and (**d**) $s = 1$ μm. It should be mentioned that the maximum number of normalized electric fields in (**b**) is 41.

## 4. Conclusions

For particle-dispersed coatings, the dependent scattering between the particles and substrate is usually ignored in applications such as radiative cooling and infrared stealth. In this work, we systematically investigated the substrate regulated mechanisms on the infrared properties of a floating particle within the atmospheric radiative window of 8–14 μm. Using the T-matrix method and open-source codes, we calculated the scattering and absorption efficiencies for the two typical cases that are encountered frequently in realistic applications. Case 1 corresponds to a polar SiO$_2$ sphere located near a metallic Al substrate, while Case 2 represents a metallic Al sphere above a polar SiO$_2$ substrate. We revealed the two main mechanisms mediating the optical properties of particle above the substrate, which were the induced charges at the small particle-substrate gaps and the standing waves for the large gaps. The intense interaction between the induced and image charges enhanced the local fields near the narrow gap and penetrated the polar SiO$_2$ materials with a small impedance. In this result, for the narrow gaps (<0.5$a$ with $a$ the sphere radius), the absorption and scattering cross sections were largely enhanced for the particles in Cases 1 and 2, respectively. For the large gaps, the interference between the particle scattered and substrate reflected waves dominates the substrate induced regulations. This interference structure (standing wave) prompts oscillating efficiency curves with the increasing gap sizes. According to a theoretical calculation, the oscillating period for the gap size was about 4.5$a$ which agreed well with the numerical results.

Furthermore, to isolate the dispersion of the optical properties of materials, we averaged the scattering and absorption efficiencies of the particles within the atmospheric

radiative window. For both Cases 1 and 2, it was found that ratio of the average efficiencies considering the substrate to those in the free space always surpassed 1 over a broad range of gap sizes (0–12*a*). We found that the dependent scattering between the particle and substrate strongly regulated the mid-infrared properties of the particles and cannot be safely ignored in practical applications. Our results may guide the accurate calculation and optimization for the apparent radiative properties of particle-dispersed coatings. In addition, the results of this work enlighten a potential regulation method for the apparent radiative property of particle-dispersed coatings by controlling the settlement distribution of the pigments.

**Author Contributions:** All authors contributed to this study's conception; methodology, F.G., S.Z. and L.M.; software, F.G.; validation, F.G., S.Z. and L.M.; investigation, F.G., S.Z. and L.M.; resources, W.Z.; data curation, F.G., S.Z. and W.Z.; writing—original draft preparation, F.G., S.Z. and W.Z.; writing—review and editing, F.G., S.Z. and W.Z.; visualization, F.G., S.Z. and W.Z.; supervision, W.Z. and L.L.; project administration, W.Z.; funding acquisition, W.Z. and S.Z. All authors have read and agreed to the published version of the manuscript.

**Funding:** This research was funded by the National Natural Science Foundation of China (Grant No. 52006127) and the Natural Science Foundation of Shandong Province (Grant Nos. ZR2022QE245 and ZR2020QE194).

**Institutional Review Board Statement:** Not applicable.

**Informed Consent Statement:** Not applicable.

**Data Availability Statement:** The data presented in this study are available on request from the corresponding author.

**Acknowledgments:** The authors would like to sincerely thank D. W. Mackowski for his guidance in using the MSTM program and his illuminating discussion on the scattering enhancement mechanism.

**Conflicts of Interest:** The authors declare no conflict of interest.

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
