# Peer review of "Significant Substrate Effects on Electromagnetic Scattering by Particles in the Infrared Atmospheric Window"

_photonics, doi:10.3390/photonics10040476_

Round 1
Reviewer 1 Report
THis paper is well written, it could be accepted in it present form.
Author Response
We appreciate your positive comments and recommendation.
Reviewer 2 Report
In this study, the authors explore the particle-substrate interactions within the atmospheric radiative window of 8-14μm. Using the T-matrix method, we calculate the scattering and absorption efficiencies of a dielectric/metallic particle situated above a metallic/dielectric substrate, considering the different gap sizes. The paper is clearly stated, and the simulation and experimental data are detailed. The paper is well-written and interesting. I recommend accepting the research after minor revisions.
i. Could the authors provide more detail on the T-matrix method used to calculate the scattering and absorption efficiencies of the particles, and how it compares to other computational methods used for this type of analysis?
ii. What are the implications of the findings for the practical design and optimization of particle-dispersed coatings, and how might these insights be applied in real-world applications?
iii. Are there any limitations or potential sources of error in the methodology or analysis that the authors have identified, and how might these affect the accuracy or generalizability of the results? The discussion on the obtained results needs to be increased for a better understanding of the reader about the significance of utilization of the proposed method.
N/A
Reviewer 3 Report
In the paper 'significant substrate effects on electromagnetic scattering by particles in the infrared atmospheric window' the authors present their recent results about scattering/absorption of sphere over different substrates. The paper is generally well-written and the results are clearly presented. However, I have few minor comments which I list here:
1. Line 13 of the abstract a space is missing 8-14um.
2. I think the authors should better comment the case of dielectric sphere over dielectric substrates as dielectric platform represent an important reasearch field in modern nanophotonics.
3. Related to 2. I think the references should be expanse by citing some fundamental works of AlGaAs nanoantennas on dielectric substrate (very used in Second Harmonic Generation / Sum Frequency Generation).
